# Genetic-Gated Networks
# for Deep Reinforcement Learning

**Simyung Chang**
Seoul National University,
Samsung Electronics
Seoul, Korea
timelighter@snu.ac.kr

**John Yang**
Seoul National University
Seoul, Korea
yjohn@snu.ac.kr

**Jaeseok Choi**
Seoul National University
Seoul, Korea
jaeseok.choi@snu.ac.kr

**Nojun Kwak**
Seoul National University
Seoul, Korea
nojunk@snu.ac.kr

## Abstract

We introduce the Genetic-Gated Networks (G2Ns), simple neural networks that combine a gate vector composed of binary genetic genes in the hidden layer(s) of networks. Our method can take both advantages of gradient-free optimization and gradient-based optimization methods, of which the former is effective for problems with multiple local minima, while the latter can quickly find local minima. In addition, multiple chromosomes can define different models, making it easy to construct multiple models and can be effectively applied to problems that require multiple models. We show that this G2N can be applied to typical reinforcement learning algorithms to achieve a large improvement in sample efficiency and performance.

## 1 Introduction

Many reinforcement learning algorithms such as policy gradient based methods [14, 17] suffer from the problem of getting stuck in local extrema. These phenomena are essentially caused by the updates of a function approximator that depend on the gradients of current policy, which is usual for on-policy methods. Exploiting the short-sighted gradients should be balanced with adequate explorations. Explorations thus should be designed irrelevant to policy gradients in order to guide the policy to unseen states. Heuristic exploration methods such as $\epsilon$-greedy action selections and entropy regularization [20] are widely used, but are incapable of complex action-planning in many environments [7, 8]. While policy gradient-based methods such as Actor-Critic models [5, 6] explore a given state space typically by applying a random Gaussian control noise in the action space, the mean and the standard deviation of the randomness remain as hyper-parameters to heuristically control the degree of exploration.

While meaningful explorations can also be achieved by applying learned noise in the parameter space and thus perturbing neural policy models [2, 9, 10], there have been genetic evolution approaches for the exploration control system of an optimal policy, considering the gradient-free methods are able to overcome confined local optima [11]. Such *et al*. [16] vectorize the weights of an elite policy network and mutate the vector with a Gaussian distribution to generate other candidate policy networks. This process is iterated until an elite parameter vector which yields the best fitness score is learned. While their method finds optimal parameters of a policy network purely based on a genetic algorithm, ES algorithm in [12] is further engaged with zero-order optimization process; the parameter vector is

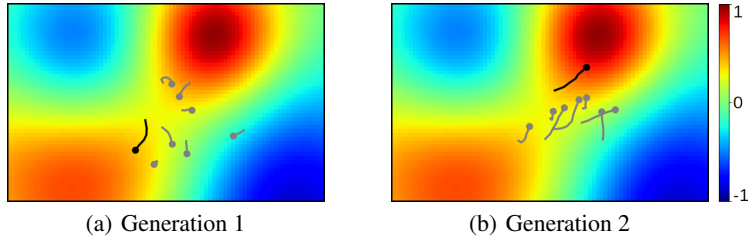

|                    |                    |
| :----------------: | :----------------: |
| (a) Generation 1   | (b) Generation 2   |

Figure 1: The optimization of Genetic-Gated Networks for a $2D$ toy problem with 8 population. The black line depicts the gradient-based training path of a network with an elite chromosome implemented. While training an elite network model, networks perturbed by other chromosomes explore other regions of the surface by using the updated network parameters trained by the elite chromosome (gray lines). All chromosomes are evaluated by their fitness scores at the end of every generation, and the training continues with the best chromosome selected.

updated every iteration with the sum of vectors that are weighted with the total score earned by their resulting policy networks. Such structure partially frees their method from back-propagation operations, which thus yields an RL algorithm to learn faster in terms of wall-clock time, but carries an issue of low sampling efficiency.

We, in this paper, introduce the Genetic-Gated Networks (G2Ns) that are composed of two phases of iterative optimization processes of gradient-free and gradient-based methods. The genetic algorithm, during the gradient-free optimization, searches for an elite chromosome that optimally perturbs an engaging neural model, hopping over regions of a loss surface. Once a population of chromosomes is generated to be implemented in a model as a dropout mask, the perturbation on the model caused by various chromosomes is evaluated based on the episodic rewards of each model. A next generation of chromosome population is generated through selections, cross-overs, and mutations of the superior gene vectors after a period of episodes. An elite model can then be selected and simply optimized further with a gradient-based learning to exploit local minimum.

A population of multiple chromosomes can quickly define unique neural models either asynchronously or synchronously, by which diverse exploration policies can be accomplished. This genetic formulation allows a model to not only explore over a multi-modal solution space as depicted in Figure 1 (The gray lines in the figures indicate multiple explorations while black line exploits the gradient), but also hop over the space whenever it finds a better region (In Figure 1(b), black is switched to one of the gray with a better solution, and then follows the gradient).

## 2    Background

**Genetic Algorithm** : A genetic algorithm [4] is a parallel and global search algorithm. It expresses the solution of the problem in the form of a specific data structure, and use a method of gradually finding better solutions through *crossovers* and *mutations*. In practice, this data structure is usually expressed as a binary vector each dimension of which decides the activation of each gene. A vector with a unique combination of these genes can be considered as a *chromosome*. Evolution begins with a population of randomly generated individuals, and every time a generation is repeated, the *fitness* of each individual is evaluated. By selecting the best chromosomes, and doing crossovers among them, better solutions are found. Through repeated selections of superior individuals and crossovers of them over generations, newly created genes in the next generation are more likely to inherit the characteristics of the superior predecessors. Additionally, the algorithm activates new genes with a fixed probability of mutations in every generation, and these random mutations allow the genetic algorithm to escape from local minima.

**Actor-Critic methods** : In actor-critic methods, an actor plays a role of learning policy $\pi_\theta(a|s_t)$ which selects action $a \in \mathcal{A}$ given that state $s$ is $s_t \in \mathcal{S}$ and a critic carries value estimation $V_w(s)$ to lead the actor to learn the optimal policy. Here, $\theta$ and $w$ respectively denote the network parameters of the actor and the critic. Training progresses towards the direction of maximizing the objective

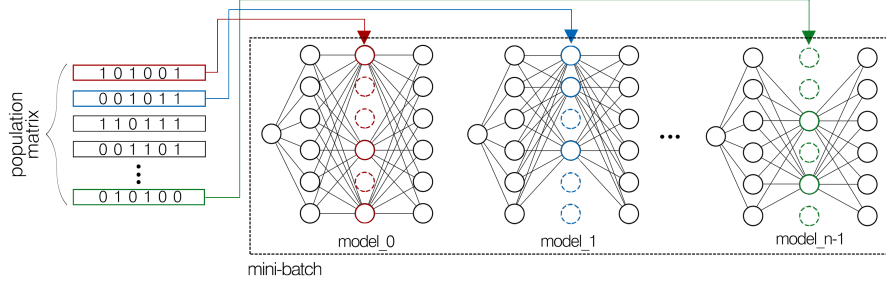

Figure 2: An illustration of a Genetic-Gated Network variously gating feed-forwarding flow of a neural network to create different models with chromosome vectors generated from a genetic algorithm.

function based on cumulative rewards, $J(\theta) = \mathbb{E}_{\pi_\theta}[\sum_t \gamma^t r_t]$ where $r_t$ is the instantaneous reward at time $t$ and $\gamma$ is a discount factor. The policy update gradient is defined as follows:

$$\nabla_\theta J(\theta) = \mathbb{E}_\pi[\nabla_\theta \log \pi_\theta(s,a) A^\pi(s,a)]. \tag{1}$$

In (1), $A^\pi(s,a)$ is an advantage function, which can be defined in various ways. In this paper, it is defined as it is in asynchronous advantage actor-critic method (A3C) [6]:

$$A^\pi(s_t, a_t) = \sum_{i=0}^{k-1} \gamma^i r(s_{t+i}, a_{t+i}) + \gamma^k V_w^\pi(s_{t+k}) - V_w^\pi(s_t), \tag{2}$$

where $k$ denotes the number of steps.

**Dropout** : Dropout [15] is one of various regularization methods for training neural networks to prevent over-fitting to training data. Typical neural networks use all neurons to feed-forward, but in a network with dropout layers, each neuron is activated with a probability $p$ and deactivated with a probability $1 - p$. Using this method, dropout layers interfere in encoding processes, causing perturbations in neural models. In that sense, each unit in a dropout layer can be interpreted to explore and re-adjust its role in relation to other activated ones. Regularizing a neural network by applying noise to its hidden units allows more robustness, enhancing generalization through training the deformed outputs to fit into the target labels. We are motivated the method can be utilized as a way to control the exploration of a model for contemporary reinforcement learning issues.

## 3 Genetic-Gated Networks

In this section, the mechanisms of Genetic-Gated Networks (G2Ns) are described and how they are implemented within the framework of actor-critic methods are explained. The proposed actor-critic methods are named as Genetic-Gated Actor-Critic (G2AC) and Genetic-Gated PPO (G2PPO).

### 3.1 Genetic-Gated networks

In Genetic-Gated networks (G2Ns), hidden units are partially gated (opened or closed) by a combination of genes (a chromosome vector) and the optimal combination of genes is learned through a genetic algorithm. The element-wise feature multiplication with a binary vector appears to be similar to that of the dropout method, yet our detailed implementation differs in the following aspects:

1. While a dropout vector is composed of Bernoulli random variables each of which has probability of $p$ being 1, a chromosome vector in G2N is generated by a genetic algorithm.

2. While a dropout vector is randomly generated for every batch, a chromosome vector stays fixed during several batches for evaluation of its fitness.

3. While the dropout method is designed for generalization in gradient-based learnings and thus performs back-propagation updates on all the 'dropped-out' models, the G2N only performs gradient-based updates on one *elite* model.

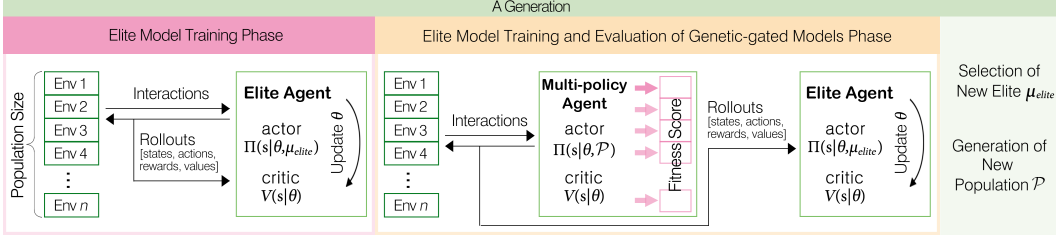

Figure 3: Interaction of multiple agents (models) and environments during GA+elite training phase. The actions are taken by multiple policy actors (multi-policy agent) with the minibatch and $\theta$ is updated using back-propagation on the elite actor with the collected information. $\mu_{elite}$ is the chromosome vector of the elite and $\mathcal{P}$ denotes the Population Matrix.

If conventional neural networks can be expressed as $f(x; \theta)$, where $x$ is an input and $\theta$ is a set of weight parameters, G2N can be defined as a function of $\theta$ and a chromosome $\mu$ which is represented as a binary vector in our paper. The perturbed model with the $i$-th genotype, $\mu_i$, can be expressed as its phenotype or an *Individual*$_i$, $f(x; \theta, \mu_i)$, and a *Population* is defined as a set of $N$ number of individuals in a generation:

$$
\begin{aligned}
Individual_i &= f(x; \theta, \mu_i), \\
Population(N) &= \{f(x; \theta, \mu_0), ..., f(x; \theta, \mu_{N-1})\}.
\end{aligned}
\tag{3}
$$

We suggest a *Population Minibatch* to train a G2N efficiently using conventional neural networks training methods. A 2D matrix called *Population Matrix*, $\mathcal{P}$, allows evaluating fitness in batches for a population of individuals. As depicted in Figure 2, since each row of this matrix is an individual chromosome, the outputs for all models of entire population can be acquired with an element-wise multiplication of this matrix and every batch. The population in (3) can thus be expressed as:

$$
Population(N) = f(x; \theta, \mathcal{P}).
\tag{4}
$$

Using the population matrix, we can evaluate fitness for multiple individuals simultaneously. This method is structurally simple, and can easily be parallelized with GPUs. Furthermore, a generation can be defined as $M$ iterations of minibatch during the learning of neural networks.

Implementations of multiple chromosomes in a parameter space as a form of gated vector enables generating various models through perturbations on the original neural model to better search over a solution space. Not only so, but every once in a generation the training model is allowed to switch its gating vector that results the best fitness-score, which comes as a crucial feature of our optimization for highly multi-modal solution spaces.

Therefore, among a population, an elite neural network model $f(x; \theta, \mu_{elite})$ is selected, where $\mu_{elite}$ denotes the chromosome that causes an individual to yield the best fitness-score. More specifically, after each generation, an elite chromosome is selected based on the following equation:

$$
\mu_{elite} = \arg\max_{\mu} F(\theta, \mu), \mu \in \{\mu_1, \ldots, \mu_{N-1}\},
\tag{5}
$$

where $F(\cdot)$ is a fitness-score function.

The procedures of 'training network parameters', 'elite selection' and 'generation of new population' are repeated during a training criterion. G2N learns the parameters $\theta$ to maximize $F$. $F$ values are then recalculated for all the individuals based on the updated parameters $\theta$ for sorting top elite chromosomes. A generation ends and a new population is generated by the genetic operations such as selections, mutations, and crossovers among chromosomes.

## 3.2 Genetic-Gated Actor Critic Methods

Genetic-Gated Actor Critic (G2AC) is a network that incorporates G2N and the conventional advantage actor critic architecture. For our back-bone model, the Synchronous Advantage Actor-Critic

**Algorithm 1** A Genetic-Gated Actor Critic Pseudo-Code

---
    Initialize Population Matrix $\mathcal{P}$
    Initialize Fitness Table $\mathcal{T}$
    $\mu_{elite} \leftarrow \mathcal{P}[0]$
    **repeat**
        Set $\mu_{elite}$ to gate matrix for acting and updating
        **while** *Elite Phase* **do**
            Do conventional Actor-Critic method
        Set $\mathcal{P}$ to gate matrix for acting
        Set $\mu_{elite}$ to gate matrix for updating
        **while** *GA+elite Phase* **do**
            Do conventional Actor-Critic method
            **if** terminated episodes exist **then**
                Store episode scores to $\mathcal{T}$
        Evaluate the fitness of the individuals using $\mathcal{T}$
        Select best individuals for the next generation
        $\mu_{elite} \leftarrow GetBestChromosome(\mathcal{P})$
        Build $\mathcal{P}$ with genetic operations such as crossovers and mutations
    **until** stop criteria

---

model (A2C) [21] which is structurally simple and publicly available. G2AC, in a later section, is evaluated in environments with discrete action space. And for continuous control, we have applied G2N in the Proximal Policy Optimization method (PPO) [13], denoting as G2PPO. Our method can be embarked on existing actor-critic models without extra weight parameters since the element-wise multiplication with the population matrix is negligible compared to the main actor-critic computations. The proposed Genetic-Gated actor critic methods (G2AC and G2PPO) have the following main features :

**Multi Model Race:** A single critic evaluates the state values while multiple agents follow unique policies, so the population matrix can be applied to the last hidden layer of actors. Therefore, our method is composed of multiple policy actors and a single critic. These multiple actors conduct exploration and compete each other. Then their fitness-scores are considered when generating new chromosomes in the following generation.

**Strong Elitism:** Our method is involved with the *strong elitism* technique. This technique not only utilizes elitism in conventional genetic algorithms which leaves elite in a preserved manner to the next generation without any crossover and mutation, but also performs back-propagations based on the model with the elite chromosome.

**Two-way Synchronous Optimization:** In every generation, our method consists of two training phases: the phase only the elite model interacts with environments to be trained and another phase that the elite and other perturbed models are co-utilized. The first phase is purposed on exploiting its policy to learn neural network parameters with relatively less explorations. Preventing value evaluations for multiple policies during a whole generation, the first phase also secures steps of training the value function with a single current policy. G2AC and G2PPO, in the second phase, use all the perturbed models to collect experiences, while updating the model parameters using the loss based on the elite model; this phase is intended to explore much more so that the elite model can be switched to the actor with a better individual if found.

Figure 3 and Algorithm 1 sum up the overall behavior of our method, being applicable for both G2AC and G2PPO. All agents interact with multiple environments and collect observations, actions and rewards into a minibatch. The elite agent is then updated on every update interval. During a generation, chromosomes are fixed and the fitness-score of each actor is evaluated. We define an actor's fitness to be the average episodic score of an actor during the generation. Figure 1 visualizes the optimization of Genetic-Gated Actor Critic in a $2D$ toy problem. It illustrate how perturbed policies (gray) behave while an elite (black) of G2N is optimized towards getting higher rewards (red), and how hopping occurs in the following generation. The detail of $2D$ toy problem and implementations are included in the supplementary material.

Table 1: The seventh column shows the average return of 10 episodes evaluated after training 200M frames from the Atari 2600 environments with G2AC(with 30 random initial no-ops). The results of DQN [19], A3C [6], HyperNeat [3], ES and A2C [12] and GA [16] are all taken from the cited papers. **Time** refers to the time spent in training, where DQN is measured with 1 GPU(K40), A3C with 16 core cpu, ES and Simpe GA with 720 CPUs, and G2AC with 1 GPU(Titan X). Hyphen marks, "-", indicate no evaluation results reported.

| FRAMES, TIME | DQN 200M, 7-10D | A3C FF 320M, 1D | HYPERNEAT - | ES 1B, 1H | SIMPLE GA 1B, 1H | G2AC 200M, 4H | A2C FF 320M, - |
|---|---|---|---|---|---|---|---|
| AMIDAR | **978.0** | 283.9 | 184.4 | 112.0 | 216 | 929.0 | 548.2 |
| ASSAULT | 4,280.4 | 3,746.1 | 912.6 | 1,673.9 | 819 | **15,147.2** | 2026.6 |
| ATLANTIS | 279,987.0 | 772,392.0 | 61,260.0 | 1,267,410.0 | 79,793 | **3,491,170.9** | 2,872,644.8 |
| BEAM RIDER | 8,627.5 | 13,235.9 | 1,412.8 | 744.0 | - | **13,262.7** | 4,438.9 |
| BREAKOUT | 385.5 | 551.6 | 2.8 | 9.5 | - | **852.9** | 368.5 |
| GRAVITAR | 473.0 | 269.5 | 370.0 | **805.0** | 462 | 665.1 | 256.2 |
| PONG | 19.5 | 11.4 | 17.4 | 21.0 | - | **23.0** | 20.8 |
| QBERT | 13,117.3 | 1,3752.3 | 695.0 | 147.5 | - | **15,836.5** | 15,758.6 |
| SEAQUEST | **5,860.6** | 2,300.2 | 716.0 | 1,390.0 | 807 | 1,772.6 | 1763.7 |
| SPACE INVADERS | 1,692.3 | 2,214.7 | 1,251.0 | 678.5 | - | **2976.4** | 951.9 |
| VENTURE | 54.0 | 19.0 | 0.0 | 760.0 | **810.0** | 0.0 | 0.0 |

# 4 Experiments

We have experimented the Atari environment and MuJoCo [18], a representative RL problems, to verify the followings: (1) Can G2N be effectively applied to Actor-Critic, a typical RL algorithm? (2) Does the genetic algorithm of Genetic-Gated RL models have advantages over a simple multi-policy model? (3) Are Genetic-Gated RL algorithms effective in terms of sample efficiency and computation? All the experiments were performed using OpenAI gym [1].

## 4.1 Performance on the Atari Domain

For the Atari experiments, we have adapted the same CNN architecture and hyper-parameters of A2C [21]. And at the beginning of training, each gene is activated with a probability of 80%. G2AC use 64 individuals (actors), with 80% of crossover probability and 3% of mutation probability for each genetic evolution. In every generation, the elite phase and the GA+elite phase are respectively set to persist 500 steps and 20 episodes for each actor.

Table 1 shows the comparable performances of G2AC on Atari game environments. When trained for far less frames than what A2C is trained, our method already outperforms the baseline model, A2C, in all the environments. Considering the performance with a little additional matrix multiplication operations, G2N proves its competency compared to many RL algorithms. In the case of GRAVITAR, even though both A3C and A2C, which are policy gradient methods, have a lower score than value-based DQN, G2AC achieves a higher score with less frames of training. However, for SEAQUEST, it seems that G2AC is not able to deviate from the local minima, scoring a similar rewards as A2C. This is presumably caused by the gate vectors that do not perturb the network enough to get out of the local minima or the low diversity of the models due to insufficient crossovers and/or mutations. For the case of VENTURE, both A2C and G2AC have not gained any reward, implying the learning model, though of an additional implementation of G2N, is constrained within the mechanism of its baseline model.

We also have conducted experiments in 50 Atari games with the same settings as those of Table 1. Compared against the base model of A2C which is trained over 320M frames, G2AC, trained only for 200M frames, has better results in 39 games, same results in three games, and worse results in the remaining eight games. Considering that it can be implemented structurally simple with a little computational overhead, applying G2N to Actor-Critic method is considered very effective in the Atari environments. The detailed experimental results are included in the supplementary material.

## 4.2 Continuous action control

We, in this section, have further experimented G2PPO in eight MuJoCo environments [18] to evaluate our method in environments with continous action control. Except that the elite phase and the

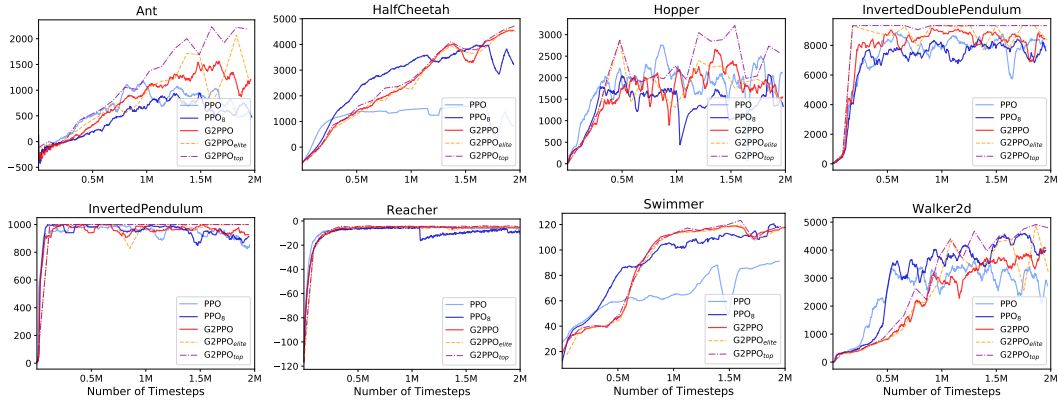

Figure 4: Performance of G2PPO on MuJoCo environments. PPO uses the same hyper parameters with its original paper while PPO$_8$ and G2PPO use eight actors (in 8 threads) and 512 steps. Compared to PPO$_8$ that uses the same setting, the proposed G2PPO is slightly lower in performance at the beginning of learning in most environments, but performs better than the baseline (PPO$_8$) as learning progresses. The score of G2PPO is the average episodic score of all the eight candidate actors while G2PPO$_{elite}$ and G2PPO$_{top}$ are the scores of the current elite and the top score among the eight actors which will be the elite in the next generation. The curve for G2PPO$_{elite}$ clearly shows the hopping property of the proposed method.

GA+elite phase are respectively set to persist for 10,240 steps and five episodes in every generation, genetic update sequence is identical to how it is in the Atari domain.

Unlike our method that is engaged with multiple actors, original PPO is reported to use a single actor model and a batch size of 2,048 steps (horizons) when learning in the MuJoCo environments. Since the number of steps $k$ is a significant hyperparameter as noted earlier in Equation 2, we have been very careful on finding right parameters of horizon size to reproduce the reported performance of PPO using multiple actor threads. We have found that PPO$_8$ can be trained synchronously in eight number of actor threads with the horizon size of 512 and reproduce most of PPO's performance in the corresponding paper. G2PPO thus has been experimented with the same settings as those of PPO$_8$ for a fair comparison. The results of three models (PPO, PPO$_8$, G2PPO) are shown in Figure 4. To monitor scores and changes of elite actors at the end of every generation, we have also indicated the scores of the current elite actor, G2PPO$_{elite}$, and the scores of a current candidate actor with the highest score, G2PPO$_{top}$, in Figure 4. Note that G2PPO$_{top}$ is always above or equal to G2PPO$_{elite}$.

As it can be seen in Figure 4, the early learning progress of G2PPO is slower than that of PPO$_8$ in most environments. Its final scores, however, are higher than or equal to the scores of the base model except for WALKER2D. The score difference between G2PPO and PPO is not significant because PPO is one of the state-of-the-art methods for many MuJoCo environments, and some environments such as InvertedDoublePendulum, InvertedPendulum and Reacher have fixed limits of maximum scores. Not knowing the global optimum, it is difficult to be sure that G2PPO can achieve significant performance gain over PPO. Instead, the superiority of G2PPO over PPO should therefore be judged based on whether G2PPO learns to gain more rewards when PPO gets stuck at a certain score despite additional learning steps as the case of Ant. For the case of WALKER2D, the result score of G2PPO have not been able to exceed the scores of baseline models when trained for 2M timesteps. Training over additional 2M steps, our method obtains 5032.8 average score of ten episodes while PPO remains at the highest score of 4493.7. Considering the graph of G2PPO follows the average episodic score of all candidate actors, the performance of G2PPO$_{elite}$ and G2PPO$_{top}$ should also be considered as much. If a single policy actor is to be examined in contrast to PPO which trains one representative actor, the performance of G2PPO$_{elite}$ and G2PPO$_{top}$ should be the direct comparison since G2PPO always selects the policy with the highest score.

Additionally, Figure 4 implies (1) if a candidate actor scores better than the current elite actor, it becomes the elite model in the next generation. This process is done as a gradient-free optimization depicted in Figure 1, hopping over various models. (2) And, if the scores of G2PPO$_{elite}$ and G2PPO$_{top}$ are equal, the current elite actor continues to be the elite. For MuJoCo environments, the changes of

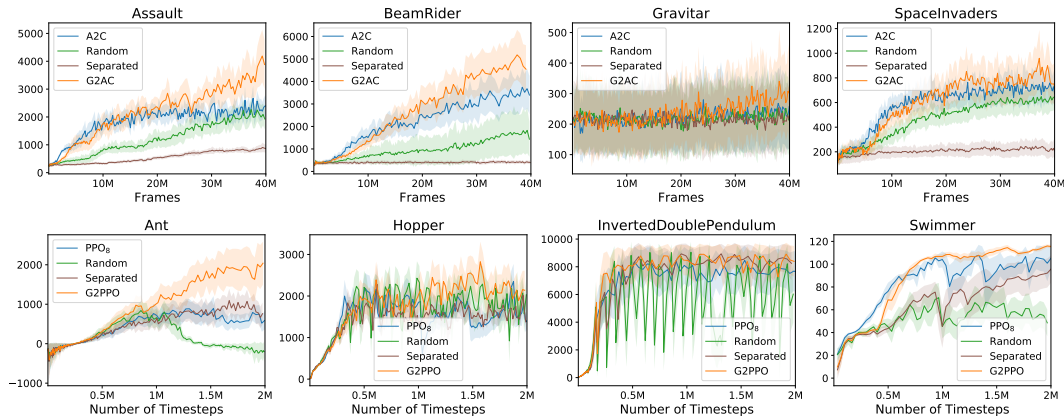

Figure 5: Performance comparisons with ablation models. *Random* denotes the multi-model with a random binary vector as gates of the same hidden layer. *Separated* denotes the model with separated optimization of GA and neural network. Each curve averaged across three trials of different random seeds. (best viewed in color)

the elite model occurred least frequently in INVERTEDPENDULUM (4 changes during 17 generations) and most frequently in REACHER (23 changes during 24 generations). The detailed results for the changes of elite model are included in supplementary material.

## 4.3   Additional Analysis

**Comparison with random multi-policy:** To compare with a simple multi-policy exploration, we have defined a multi-model structure with random binary vectors as gates and have compared the performance of it with our methods. The other conditions, such as the generation period and the number of models, are set to be the same as G2AC and G2PPO. Random binary vectors are generated with the same probability as genes initial probability and used instead of chromosomes. By doing so, we have obtained a multi-policy model that does not use a genetic algorithm.

As shown in the Figure 5, using a multi-policy model with random gates does not bring a significant improvement over the baseline and even results in worse performance in a few environments. The performance reductions are especially observed in experiments with the MuJoCo environments. This is due to the direct perturbation of the continuous action, which causes highly unstable results as the gate vector changes. On the other hand, the G2N models have achieved noticeable performance improvements. In Atari environments, G2AC with 40M frames of training has a higher average score than A2C with 320M learning frames. G2PPO in MuJoCo experiments is shown to improve performances by effectively escaping local extrema. Furthermore, the learning curves of G2AC and G2PPO in Fig. 5 are drawn based on the average scores of all perturbed agents, not the best ones.

**Two-way synchronous optimization:** Gradient-free optimization methods are generally known to be slower than gradient-based optimizations. ES and GA algorithms have achieved comparable performances after training 1B frames which are much larger number than those of DQN, A3C and A2C with traditional deep neural networks. *Separated* in Figure 5 denotes learning curves of the separated optimization of GA and its neural model. This allows us to compare the efficiency with the two-way synchronous optimization. The graph clearly shows learning in two-way synchronous methods (G2AC, G2PPO) are much faster. The gap is larger in Atari, which has relatively high proportion of Genetic+elite phase. These results show that the sampling efficiency can be improved by training neural networks while evaluating the fitness of individuals rather than pausing the training.

**Wall-clock time:** As described earlier, G2AC does not require additional operations other than element-wise multiplication of a batch and the gate matrix and creating a new population at the end of each generation. Experiments on five Atari games have shown that G2AC takes 3572 steps per second while A2C takes 3668. G2PPO which operates in parallel with multiple actors like $PPO_8$ completes 1110 steps per second while $PPO_8$ and PPO complete 1143 and 563 steps respectively. Our methods were slowed within only 3% when compared to their direct baselines.

**Hyper-parameters of Genetic algorithm:** We do not aim to find the highest performance model by tuning the hyper-parameters of the genetic algorithm. However, we have experimented with some hyper-parameters to see what characteristics G2N shows for these. G2AC is less sensitive to the hyper-parameters of the genetic algorithm while G2PPO is more sensitive. This is as anticipated considering that G2AC uses softmax activation as its output, so the range of the perturbed output is limited. On the other hand, the outputs of G2PPO are boundless and therefore they are directly affected by the binary gates. In MuJoCo, as the mutation probability increases from 0.03 to 0.1 and 0.3, the performance decreases and becomes unstable. In the case of crossover probability, the difference is higher in Hopper and Swimmer when changing from 0.8 to 0.4. But, the influence of crossover was not significant in other environments. The detailed results of the experiment are included in the supplementary material.

## 5 Conclusions

We have proposed a newly defined network model, Genetic-Gated Networks (G2Ns). G2Ns can define multiple models without increasing the amount of computation and use the advantage of gradient-free optimization by simply gating with a chromosome based on a genetic algorithm in a hidden layer. Incorporated with the genetic gradient-based optimization techniques, this gradient-free optimization method is expected to find global optima effectively for multi-modal environments.

As applications of G2Ns, we have also proposed Genetic-Gated Actor-Critic methods(G2AC, G2PPO) which can be used to problems within the RL domain where the multiple models can be useful while having the local minima problem. Our experiments show that the performance of the base model is greatly improved by the proposed method. It is not just a development of a RL algorithm, but it shows that a combination of two completely different machine learning algorithms can be a better algorithm.

In future works, we intend to study whether G2Ns can be applied to other domains that are related with multiple local minima or can benefit from using multiple models. Also, we need to study how to overcome the initial learning slowdown due to the additional exploration of perturbed polices.

### Acknowledgement

This work was supported by Next-Generation Information Computing Development Program through the National Research Foundation of Korea (NRF) (2017M3C4A7077582).

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
