[Supplementary Material]

# Supplementary Material – Genetic-Gated Networks for Deep Reinforcement Learning

**Simyung Chang**
Seoul National University,
Samsung Electronics
Seoul, Korea
timelighter@snu.ac.kr

**John Yang**
Seoul National University
Seoul, Korea
yjohn@snu.ac.kr

**Jaeseok Choi**
Seoul National University
Seoul, Korea
jaeseok.choi@snu.ac.kr

**Nojun Kwak**
Seoul National University
Seoul, Korea
nojunk@snu.ac.kr

## A  Experiment Settings

### A.1  Atari Environment

For the Atari environment experiments, we adapted the same CNN architecture (size 3x3 with stride 1) with DQN, ACKTR and A2C [6, 7]. And then, a fully connected layer of size 512 is connected followed by a genetic gate vector with the same size. Finally it produces action probabilities using fully connected layer with softmax activation. We used ReLU activation for all layers except the output layer.

| Hyperparameter | G2AC |
|---|---|
| Optimizer | RMSProp |
| learning rate | 0.0007 |
| Epsilon ($\epsilon$) | 0.00001 |
| Alpha ($\alpha$) | 0.99 |
| Number of actors | 64 |
| Elite phase (steps per actor) | 500 |
| GA+Elite phase (episodes) | 20 |
| Keep probability | 0.8 |
| Mutation probability | 0.03 |

Table 1: G2AC hyperparameters used for Atari experiments.

## A.2 MuJoCo Environment

| HYPERPARAMETER | PPO | PPO$_8$ | G2PPO |
|---|---|---|---|
| HORIZON (T) | 2048 | 512 | 512 |
| ADAM STEPSIZE | $3 \times 10^{-4}$ | $3 \times 10^{-4}$ | $3 \times 10^{-4}$ |
| NUM. EPOCHS | 10 | 10 | 10 |
| MINIBATCH SIZE | 64 | 64 | 64 |
| DISCOUNT ($\gamma$) | 0.99 | 0.99 | 0.99 |
| GAE PARAMETER ($\lambda$) | 0.95 | 0.95 | 0.95 |
| NUMBER OF ACTORS | 1 | 8 | 8 |
| ELITE PHASE (STEPS PER ACTOR) | - | - | 10,240 |
| GA+ELITE PHASE (EPISODES) | - | - | 5 |
| KEEP PROBABILITY | - | - | 0.8 |
| MUTATION PROBABILITY | - | - | 0.03 |

Table 2: PPO, PPO$_8$ and G2PPO hyperparameters used for MuJoCo experiments. PPO [5] hyperparameters are same with the cited paper.

## A.3 $2D$ Toy Problem

To visualize the optimization of G2N, we have used the $2D$ Toy Problem of the blog [1] which OpenAI create to illustrate their Evolution strategy [4]. The $2D$ Toy Problem is an environment that gives a high or low reward for the x, y coordinates as shown in Figure 1 of our paper. G2N, based on the G2PPO used in the MuJoCo environment, outputs the x and y coordinates and is trained with rewards given in the environment. In order to clarify the effect of the perturbed policy according to the elite update, the agent use only the collected from the elite, rather than the experiences collected from the multi-policy agent. The detailed hyper parameters are as follows.

| HYPERPARAMETER | G2AC |
|---|---|
| HORIZON (T) | 32 |
| ADAM STEPSIZE | $1 \times 10^{-4}$ |
| NUM. EPOCHS | 2 |
| MINIBATCH SIZE | 16 |
| DISCOUNT ($\gamma$) | 0.99 |
| GAE PARAMETER ($\lambda$) | 0.95 |
| NUMBER OF ACTORS | 8 |
| ELITE PHASE (STEPS PER ACTOR) | 0 |
| GA+ELITE PHASE (EPISODES) | 200 |
| KEEP PROBABILITY | 0.3 |
| MUTATION PROBABILITY | 0.03 |

Table 3: G2N hyperparameters used for $2D$ Toy Problem.

# B  Additional Experimental Results

## B.1  50 Atari games

| FRAMES, TIME | DQN 200M, 7-10D | A3C FF 320M, 1D | HYPERNEAT - | ES 1B, 1H | A2C FF 320M, - | G2AC 200M, 4H |
|---|---|---|---|---|---|---|
| AMIDAR | 133.40 | 283.90 | 184.40 | 112.00 | 548.20 | **929.00** |
| ASSAULT | 3,332.30 | 3,746.10 | 912.60 | 1,673.90 | 2,026.60 | **15,147.20** |
| ASTERIX | 124.50 | 6,723.00 | 2,340.00 | 1,440.00 | 3,779.70 | **67,972.46** |
| ASTEROIDS | 697.10 | 3,009.40 | 1,694.00 | 1,562.00 | 1,733.40 | **4,142.45** |
| ATLANTIS | 76,108.00 | 772,392.00 | 61,260.00 | 1,267,410.00 | 2,872,644.80 | **3,491,170.90** |
| BANK HEIST | 176.30 | 946.00 | 214.00 | 225.00 | 724.10 | **1,250.49** |
| BATTLE ZONE | 17,560.00 | 11,340.00 | **36,200.00** | 16,600.00 | 8,406.20 | 10,874.86 |
| BEAM RIDER | 8,672.40 | 13,235.90 | 1,412.80 | 744.00 | 4,438.90 | **13,262.70** |
| BERZERK | | **1,433.40** | 1,394.00 | 686.00 | 720.60 | 1,101.86 |
| BOWLING | 41.20 | 36.20 | **135.80** | 30.00 | 28.90 | 28.10 |
| BOXING | 25.80 | 33.70 | 16.40 | 49.80 | **95.80** | 7.20 |
| BREAKOUT | 303.90 | 551.60 | 2.80 | 9.50 | 368.50 | **852.90** |
| CENTIPEDE | 3,773.10 | 3,306.50 | **25,275.20** | 7,783.90 | 2,773.30 | 9,635.43 |
| CHOPPER COMMAND | 3,046.00 | **4,669.00** | 3,960.00 | 3,710.00 | 1,700.00 | 4559.22 |
| CRAZY CLIMBER | 50,992.00 | 101,624.00 | 0.00 | 26,430.00 | 100,034.40 | **167,287.03** |
| DEMON ATTACK | 12,835.20 | 84,997.50 | 14,620.00 | 1,166.50 | 23,657.70 | **458,295.90** |
| DOUBLE DUNK | **21.60** | 0.10 | 2.00 | 0.20 | 3.20 | 0.00 |
| ENDURO | **475.60** | 82.20 | 93.60 | 95.00 | 0.00 | 0.00 |
| FISHING DERBY | 2.30 | 13.60 | **49.80** | 49.00 | 33.90 | 44.80 |
| FREEWAY | 25.80 | 0.10 | 29.00 | **31.00** | 0.00 | 0.10 |
| FROSTBITE | 157.40 | 180.10 | **2,260.00** | 370.00 | 266.60 | 323.00 |
| GOPHER | 2,731.80 | 8,442.80 | 364.00 | 582.00 | 6,266.20 | **72,243.19** |
| GRAVITAR | 216.50 | 269.50 | 370.00 | **805.00** | 256.20 | 665.10 |
| ICE HOCKEY | 3.80 | 4.70 | **10.60** | 4.10 | 4.90 | -3.60 |
| KANGAROO | 2,696.00 | 106.00 | 800.00 | **11,200.00** | 1,357.60 | 140.00 |
| KRULL | 3,864.00 | 8,066.60 | **12,601.40** | 8,647.20 | 6,411.50 | 9,404.45 |
| MONTEZUMAÂS REVENGE | 50.00 | **53.00** | 0.00 | 0.00 | 0.00 | 0.00 |
| NAME THIS GAME | 5,439.90 | 5,614.00 | 6,742.00 | 4,503.00 | 5,532.80 | **13,208.74** |
| PHOENIX | | 28,181.80 | 1,762.00 | 4,041.00 | 14,104.70 | **92,939.71** |
| PITFALL | | **123.00** | 0.00 | 0.00 | 8.20 | 0.00 |
| PONG | 16.20 | 11.40 | 17.40 | 21.00 | 20.80 | **23.00** |
| PRIVATE EYE | 298.20 | 194.40 | **10,747.40** | 100.00 | 100.00 | 1,489.99 |
| Q*BERT | 4,589.80 | 13,752.30 | 695.00 | 147.50 | 15,758.60 | **15,836.50** |
| RIVERRAID | 4,065.30 | 10,001.20 | 2,616.00 | 5,009.00 | 9,856.90 | **17,805.36** |
| ROAD RUNNER | 9,264.00 | 31,769.00 | 3,220.00 | 16,590.00 | 33,846.90 | **42,763.07** |
| ROBOTANK | **58.50** | 2.30 | 43.80 | 11.90 | 2.20 | 12.57 |
| SEAQUEST | **2,793.90** | 2,300.20 | 716.00 | 1,390.00 | 1,763.70 | 1,772.60 |
| SKIING | | 13,700.00 | 7,983.60 | **15,442.50** | 15,245.80 | -8,990.11 |
| SOLARIS | | 1,884.80 | 160.00 | 2,090.00 | 2,265.00 | **3,361.74** |
| SPACE INVADERS | 1,449.70 | 2,214.70 | 1,251.00 | 678.50 | 951.90 | **2,976.40** |
| STAR GUNNER | 34,081.00 | 64,393.00 | 2,720.00 | 1,470.00 | 40,065.60 | **98,100.35** |
| TENNIS | 2.30 | 10.20 | 0.00 | 4.50 | **11.20** | -21.00 |
| TIME PILOT | 5,640.00 | 5,825.00 | **7,340.00** | 4,970.00 | 4,637.50 | 6,757.19 |
| TUTANKHAM | 32.40 | 26.10 | 23.60 | 130.30 | 194.30 | **328.57** |
| UP AND DOWN | 3,311.30 | 54,525.40 | 43,734.00 | 67,974.00 | 75,785.90 | **82,383.38** |
| VENTURE | 54.00 | 19.00 | 0.00 | **760.00** | 0.00 | 0.00 |
| VIDEO PINBALL | 20,228.10 | 185,852.60 | 0.00 | 22,834.80 | 46,470.10 | **605,324.07** |
| WIZARD OF WOR | 246.00 | 5,278.00 | 3,360.00 | 3,480.00 | 1,587.50 | **7,801.36** |
| YARS REVENGE | | 7,270.80 | **24,096.40** | 16,401.70 | 8,963.50 | 11,124.39 |
| ZAXXON | 831.00 | 2,659.00 | 3,000.00 | 6,380.00 | 5.60 | **10,922.10** |

Table 4: The last column shows the average return of 10 episodes evaluated after training 200M frames from the Atari 2600 environments with G2AC (with 30 random initial no-ops, like ES or A2C). The results of DQN [6], A3C [3], HyperNeat [2], ES and A2C [4] are all taken from the cited papers. Since A3C only reports raw scores with the human starts condition, it is difficult to compare the scores directly. **Time** refers to the time spent in training, where DQN is measured with 1 GPU (K40), A3C with 16 core CPU, ES and Simple GA with 720 CPUs, and G2AC with 1 GPU (Titan X). The - marked data was not included because there was no evaluation result in the paper cited.

## B.2 The Number of Generations and Changes of The Elite

| ENVIRONMENT | #GENERATIONS | #CHANGES OF ELITE |
|---|---|---|
| AMIDAR | 18 | 18 |
| ASSAULT | 19 | 18 |
| ATLANTIS | 12 | 11 |
| BEAM RIDER | 8 | 8 |
| BREAKOUT | 26 | 24 |
| GRAVITAR | 51 | 49 |
| PONG | 4 | 3 |
| Q*BERT | 44 | 42 |
| SEAQUEST | 14 | 14 |
| SPACE INVADERS | 20 | 18 |
| VENTURE | 14 | 0 |
| ANT | 19 | 16 |
| HALFCHEETAH | 17 | 14 |
| HOPPER | 21 | 19 |
| INVERTEDDOUBLEPENDULUM | 18 | 12 |
| INVERTEDPENDULUM | 17 | 4 |
| REACHER | 24 | 23 |
| SWIMMER | 17 | 14 |
| WALKER2D | 19 | 15 |

Table 5: The number of generations and elite changes during training. The top rows are the results of Atari and the bottom rows are the results of MuJoCo. Each numbers has been measured for 40 million frames(10 million timesteps) in Atari and 2 million timesteps in MuJoCo.

## B.3 Hyper-parameters of Genetic algorithm

Figure 1: Performance according to different hyper-parameters of genetic algorithm. *G2AC* and *G2PPO* denote the models of the paper with the mutation probability of 0.03 and crossover probability of 0.8. *mu 0.1 and mu0.3* denote the models with the mutation probability 0.1 and 0.3. *cr0.4* denotes the model with the crossover probability of 0.4. Each curve averaged across three trials of different random seeds. (best viewed in color)