[Reviews · NeurIPS 2018]

Reviewer 1



The authors propose a new RL framework that combines gradient-free genetic algorithms with gradient based optimization (policy gradients). The idea is to parameterize an ensemble of actors by using a binary gating mechanism, similar to dropout, between hidden layers. Instead of sampling a new gate pattern at every iteration, as in dropout, each gate is viewed as a gene and the activation pattern as a chromosome. This allows learning the policy with a combination of a genetic algorithm and policy gradients. The authors apply the proposed algorithm to Atari domain, and the results demonstrate significant improvement over standard algorithms. They also apply their method to continuous control (OpenAI gym MuCoJo benchmarks) yielding results that are comparable to standard PPO. In general, the idea of combining gradient-free and gradient-based algorithms is interesting as it can benefit from the fast convergence of gradient-based methods while avoiding the problem of converging to bad local minima. I’m guessing that the gym benchmarks do not suffer from bad local minima that much making the improvement over standard PPO less prominent. What slightly confuses me is how the experience is used to update the actors. As far as I understand, in the first phase, the elite actor is learned following a standard training procedure (either A2C or PPO), and in the second phase, all actors are used to collect experience, which is used to evaluate the fitness of each actor and also to train the elite actor. If that is the case, then how are the off-policy samples collected in the second phase incorporated in the elite updates? My main concern is that the experimental evaluation does not clearly answer the question what makes the method = work so well in the Atari domain. The ablations in Figure 5 indicate that, compared to standard A2C, G2AC learns initially equally well. However, the ablations are run only for the first 40M steps, and do not show what happens later during training. I’m assuming standard A2C gets stuck in a local minimum, however this cannot be inferred from the figure. Improved ablation study and inclusion of a toy problem to pinpoint what makes the proposed method work well would make the submission substantially better. --- I have read the authors response, and I have revised my score accordingly. However, I still think the experiments in continuous domains are flawed which can partially explain why the improvements are marginal. For example, state-of-the-art model-free off-policy algorithm can obtain a return of 4k ... 6k in 2M steps on Ant (see TD3, SAC), so the results reported in the paper are not close to global optimum.

Reviewer 2



The paper presents a reinforcement learning approach that has both gradient-based and gradient-free policy search nature. My understanding of the proposed algorithm is that it alternates between two stages. In the first stage, an elite policy is trained using typical RL algorithms such as PPO or A2C. Then during the second stage, a set of perturbed policies is used to generate the rollouts and update the elite policy, again with PPO or A2C. The rollouts collected during the second stage are used to estimate the performance of the perturbed policies and the best one is picked as the elite model in the next iteration. They demonstrated improved performance in playing atari games and limited improvements on continuous control problems. I think the interesting part of the proposed method is that samples from a set of perturbed policies are used not only to compute the policy gradient, but also to estimate the individual performances for a possible jump in the parameter space. However, I think the experiment part needs notable improvements, as discussed in the itemized comments below. 1. A more detailed ablation study would be greatly appreciated. For example, is the algorithm sensitive to the crossover and mutation probability? Is the elite stage (first stage) necessary at all? What if we don’t select the new elite model from the population in stage two? These would help understand which components contributes the most to the performance boost. 2. The random multi policy baseline is not very clear to me. Does it mean randomly picking an elite policy during the GA+elite phase, or entirely randomize the population at every iteration instead of using mutation and crossover? 3. What does PPO_8 mean? Is it training 8 separate policies, or one policies with 8 threads to generate rollouts? 4. How many random seeds were used for the results in figure 4? 5. Also, it would be interesting to see a comparison/discussion with other forms of policy perturbations such as gaussian noise other than binary perturbation. In general, the paper presents an interesting idea and show good improvement on playing atari games and limited improvement on continuous control problems. Some clarifications about the method and experiments are needed as discussed above and it would benefit from more experiments. ------ I have modified the score after reading the authors' response. The ablation study helps demonstrate the strength of the proposed method. However, using single random seed for the experiments in Figure 4 is not well justified and should be explicitly stated to avoid confusion.

Reviewer 3



The authors proposed the Genetic-Gated Networks (G2Ns) which combine conventional gradient based policy optimization and genetic algorithms. The gates are binary masks injected to the conventional policy neural networks, and their values are adapted via genetic algorithms while learning the policies. Specifically, a population of differently configured gene vectors are initially generated, and the elite gene configuration which has the highest fitness score is selected in order to update the conventional neural network parameters via gradient based policy optimization. The gate values are updated via genetic algorithms periodically as well. The G2Ns are applied to the actor policy of Advantage Actor-Critic (G2AC) on Atari 2600 games and Proximal Policy Optimization (G2PPO) on eight MuJoCo environments. The empirical results show that G2AC achieved improved results on 39 out of 50 Atari games and G2PPO achieved improvement on some MuJoCo environments as well. The G2Ns are general neural network framework which could be applied to either supervised learning or reinforcement learning. The reinforcement learning related aspect of G2N is in the Two-way Synchronous Optimization part. In the second phase of the two-way synchronous optimization, multiple policies with different gene configuration are leveraged to collect training trajectories. The empirical ablation studies on the two-way synchronous optimization in section 4.3 are very helpful in justifying the positive impact of such approach. The multiple policy-based exploration of G2Ns is related to the ideas of exploration via bootstrapped DQN or noisy networks. An in-depth analysis would also be helpful to highlight the uniqueness of G2Ns on exploration in general. The G2Ns achieve significant improvement on discrete action problems as it is shown that the final performance on 39 out of 50 Atari games are improved. Some sample plots of learning curves on the Atari games would also be helpful in checking that they do not suffer from the poor initial performance issue as reported in the continuous domains. Given the plots in Figure 5 on the four Atari games, it seems that G2AC is outperformed by A2C in the first 10M training, which suggests the poor initial performance issue may be an inherent issue of G2Ns, not specific to the continuous reinforcement learning domains. Any additional material on this matter would be helpful. The improvement of G2PPO in continuous domains is less significant compared to the G2AC in discrete domains as most of the curves are mixed in Figure 4. It is curious to know the hyper-parameter sensitivity of G2Ns to the initial activation probability, the crossover probability and the mutation probability. The paper is not related to multiagent RL at all. The term “multi-agent” is misused. ---- I have read the reviews and the rebuttal. The rebuttal did not change my assessment of the paper. I would like to keep my score.